# Stochastic Gradient/Mirror Descent: Minimax Optimality and Implicit Regularization

**Navid Azizan**
California Institute of Technology
Pasadena, CA 91125
`azizan@caltech.edu`

**Babak Hassibi**
California Institute of Technology
Pasadena, CA 91125
`hassibi@caltech.edu`

## Abstract

Stochastic descent methods (of the gradient and mirror varieties) have become increasingly popular in optimization. In fact, it is now widely recognized that the success of deep learning is not only due to the special deep architecture of the models, but also due to the behavior of the stochastic descent methods used, which play a key role in reaching "good" solutions that generalize well to unseen data. In an attempt to shed some light on why this is the case, we revisit some minimax properties of stochastic gradient descent (SGD) for the square loss of linear models—originally developed in the 1990's—and extend them to *general* stochastic mirror descent (SMD) algorithms for *general* loss functions and *nonlinear* models. In particular, we show that there is a fundamental identity which holds for SMD (and SGD) under very general conditions, and which implies the minimax optimality of SMD (and SGD) for sufficiently small step size, and for a general class of loss functions and general nonlinear models. We further show that this identity can be used to naturally establish other properties of SMD (and SGD), namely convergence and *implicit regularization* for over-parameterized linear models (in what is now being called the "interpolating regime"), some of which have been shown in certain cases in prior literature. We also argue how this identity can be used in the so-called "highly over-parameterized" nonlinear setting (where the number of parameters far exceeds the number of data points) to provide insights into why SMD (and SGD) may have similar convergence and implicit regularization properties for deep learning.

## 1 Introduction

Deep learning has proven to be extremely successful in a wide variety of tasks (Krizhevsky et al., 2012; LeCun et al., 2015; Mnih et al., 2015; Silver et al., 2016; Wu et al., 2016). Despite its tremendous success, the reasons behind the good generalization properties of these methods to unseen data is not fully understood (and, arguably, remains somewhat of a mystery to this day). Initially, this success was mostly attributed to the special deep architecture of these models. However, in the past few years, it has been widely noted that the architecture is only part of the story, and, in fact, the optimization algorithms used to train these models, typically stochastic gradient descent (SGD) and its variants, play a key role in learning parameters that generalize well.

In particular, it has been observed that since these deep models are *highly over-parameterized*, they have a lot of capacity, and can fit to virtually any (even random) set of data points (Zhang et al., 2016). In other words, highly over-parameterized models can "interpolate" the data, so much so that this regime has been called the "interpolating regime" (Ma et al., 2018). In fact, on a given dataset, the loss function often has (uncountably infinitely) many *global* minima, which can have drastically different generalization properties, and it is not hard to construct "trivial" global minima that do not generalize. Which minimum among all the possible minima we pick in practice is determined by the optimization algorithm that we use for training the model. Even though it may seem at first that, because of the non-convexity of the loss function, the stochastic descent algorithms may get stuck in local minima or saddle points, in practice they almost always achieve a global minimum (Kawaguchi, 2016; Zhang et al., 2016; Lee et al., 2016), which perhaps can also be justified by the fact that these models are highly over-parameterized. What is even more interesting is that not only

do these stochastic descent algorithms converge to global minima, but they converge to "special" ones that generalize well, even in the absence of any explicit regularization or early stopping (Zhang et al., 2016). Furthermore, it has been observed that even among the common optimization algorithms, namely SGD or its variants (AdaGrad (Duchi et al., 2011), RMSProp (Tieleman & Hinton, 2012), Adam (Kingma & Ba, 2014), etc.), there is a discrepancy in the solutions achieved by different algorithms and their generalization capabilities (Wilson et al., 2017), which again highlights the important role of the optimization algorithm in generalization.

There have been many attempts in recent years to explain the behavior and properties of these stochastic optimization algorithms, and many interesting insights have been obtained (Achille & Soatto, 2017; Chaudhari & Soatto, 2018; Shwartz-Ziv & Tishby, 2017; Soltanolkotabi et al., 2017). In particular, it has been argued that the optimization algorithms perform an *implicit regularization* (Neyshabur et al., 2017; Ma et al., 2017; Gunasekar et al., 2017; 2018a; Soudry et al., 2017; Gunasekar et al., 2018b) while optimizing the loss function, which is perhaps why the solution generalizes well. Despite this recent progress, most results explaining the behavior of the optimization algorithm, even for SGD, are limited to linear or very simplistic models. Therefore, a general characterization of the behavior of stochastic descent algorithms for more general models would be of great interest.

## 1.1 OUR CONTRIBUTION

In this paper, we present an alternative explanation of the behavior of SGD, and more generally, the stochastic mirror descent (SMD) family of algorithms, which includes SGD as a special case. We do so by obtaining a fundamental identity for such algorithms (see Lemmas 2 and 5). Using these identities, we show that for general nonlinear models and general loss functions, when the step size is sufficiently small, SMD (and therefore also SGD) is the optimal solution of a certain minimax filtering (or online learning) problem. The minimax formulation is inspired by, and rooted, in $H^\infty$ filtering theory, which was originally developed in the 1990's in the context of robust control theory (Hassibi et al., 1999; Simon, 2006; Hassibi et al., 1996), and we generalize several results from this literature, e.g., (Hassibi et al., 1994; Kivinen et al., 2006). Furthermore, we show that many properties recently proven in the learning/optimization literature, such as the implicit regularization of SMD in the over-parameterized linear case—when convergence happens—(Gunasekar et al., 2018a), naturally follow from this theory. The theory also allows us to establish new results, such as the convergence (in a deterministic sense) of SMD in the over-parameterized linear case. We also use the theory developed in this paper to provide some speculative arguments into why SMD (and SGD) may have similar convergence and implicit regularization properties in the so-called "highly over-parameterized" nonlinear setting (where the number of parameters far exceeds the number of data points) common to deep learning.

In an attempt to make the paper easier to follow, we first describe the main ideas and results in a simpler setting, namely, SGD on the square loss of linear models, in Section 3, and mention the connections to $H^\infty$ theory. The full results, for SMD on a general class of loss functions and for general nonlinear models, are presented in Section 4. We demonstrate some implications of this theory, such as deterministic convergence and implicit regularization, in Section 5, and we finally conclude with some remarks in Section 6. Most of the formal proofs are relegated to the appendix.

## 2 PRELIMINARIES

Denote the training dataset by $\{(x_i, y_i) : i = 1, \ldots, n\}$, where $x_i \in \mathbb{R}^d$ are the inputs, and $y_i \in \mathbb{R}$ are the labels. We assume that the data is generated through a (possibly nonlinear) model $f_i(w) = f(x_i, w)$ with some parameter vector $w \in \mathbb{R}^m$, plus some noise $v_i$, i.e., $y_i = f(x_i, w) + v_i$ for $i = 1, \ldots, n$. The noise can be due to actual measurement error, or it can be due to modeling error (if the model $f(x_i, \cdot)$ is not rich enough to fully represent the data), or it can be a combination of both. As a result, we do not make any assumptions on the noise (such as stationarity, whiteness, Gaussianity, etc.).

Since typical deep models have a lot of capacity and are highly over-parameterized, we are particularly interested in the over-parameterized (so-caled interpolating) regime, i.e., when $m > n$. In this case, there are many parameter vectors $w$ (in fact, uncountably infinitely many) that are consistent

with the observations. We denote the set of these parameter vectors by

$$\mathcal{W} = \{w \in \mathbb{R}^m \mid y_i = f(x_i, w), \ i = 1, \ldots, n\}. \tag{1}$$

(Note the absence of the noise term, since in this regime we can fully interpolate the data.) The set $\mathcal{W}$ is typically an $(m - n)$-dimensional manifold and depends only on the training data $\{(x_i, y_i) : i = 1, \ldots, n\}$ and nonlinear model $f(\cdot, \cdot)$.

The total loss on the training set (empirical risk) can be denoted by $L(w) = \sum_{i=1}^n L_i(w)$, where $L_i(\cdot)$ is the loss on the individual data point $i$. We assume that the loss $L_i(\cdot)$ depends only on the residual, i.e., the difference between the prediction and the true label. In other words,

$$L_i(w) = l(y_i - f(x_i, w)), \tag{2}$$

where $l(\cdot)$ can be any nonnegative differentiable function with $l(0) = 0$. Typical examples of $l(\cdot)$ include square ($l_2$) loss, Huber loss, etc. We remark that, in the interpolating regime, every parameter vector in the set $\mathcal{W}$ renders each individual loss zero, i.e., $L_i(w) = 0$, for all $w \in \mathcal{W}$.

## 3   WARM-UP: REVISITING SGD ON SQUARE LOSS OF LINEAR MODELS

In this section, we describe the main ideas and results in a simple setting, i.e., stochastic gradient descent (SGD) for the square loss of a linear model, and we revisit some of the results from $H^\infty$ theory (Hassibi et al., 1999; Simon, 2006). In this case, the data model is $y_i = x_i^T w + v_i$, $i = 1, \ldots, n$ (where there is no assumption on $v_i$) and the loss function is $L_i(w) = \frac{1}{2}(y_i - x_i^T w)^2$.

Assuming the data is indexed randomly, the SGD updates are defined as $w_i = w_{i-1} - \eta \nabla L_i(w_{i-1})$, where $\eta > 0$ is the step size or learning rate.[1] The update in this case can be expressed as

$$w_i = w_{i-1} + \eta \left(y_i - x_i^T w_{i-1}\right) x_i, \tag{3}$$

for $i \geq 1$ (for $i > n$, we can either cycle through the data, or select them at random).

**Remark.** We should point out that, when the step size $\eta$ is fixed, the SGD recursions have no hope of converging, unless there exists a weight vector $w$ which perfectly interpolates the data $\{(x_i, y_i) : i = 1, \ldots, n\}$. The reason being that, if this is not the case, for any estimated weight vector in SGD there will exist at least one data point that has a nonzero instantaneous gradient and that will therefore move the estimate by a non-vanishing amount.[2] It is for this reason that the results on the convergence of SGD and SMD (Sections 3.3 and 5) pertain to the interpolating regime.

### 3.1   CONSERVATION OF UNCERTAINTY

Prior to the $i$-th step of any optimization algorithm, we have two sources of uncertainty: our uncertainty about the unknown parameter vector $w$, which we can represent by $w - w_{i-1}$, and our uncertainty about the $i$-th data point $(x_i, y_i)$, which we can represent by the noise $v_i$. After the $i$-th step, the uncertainty about $w$ is transformed to $w - w_i$. But what about the uncertainty in $v_i$? What is it transformed to? In fact, we will view any optimization algorithm as one which redistributes the uncertainties at time $i - 1$ to new uncertainties at time $i$. The two uncertainties, or error terms, we will consider are $e_i$ and $e_{p,i}$, defined as follows.

$$e_i := y_i - x_i^T w_{i-1}, \text{ and } e_{p,i} := x_i^T w - x_i^T w_{i-1}. \tag{4}$$

$e_i$ is often referred to as the *innvovations* and is the error in predicting $y_i$, given the input $x_i$. $e_{p,i}$ is sometimes called the *prediction error*, since it is the error in predicting the noiseless output $x_i^T w$, i.e., in predicting what the best output of the model is. In the absence of noise, $e_i$ and $e_{p,i}$ coincide.

One can show that SGD transforms the uncertainties in the fashion specified by the following lemma, which was first noted in (Hassibi et al., 1996).

---

[1] For the sake of simplicity of presentation, we present the results for constant step size. We show in the appendix that all the results extend to the case of time-varying step-size.

[2] Of course, one may get convergence by having a vanishing step size $\eta_i \to 0$. However, in this case, convergence is not surprising—since, effectively, after a while the weights are no longer being updated—and the more interesting question is "what" the recursion converges to.

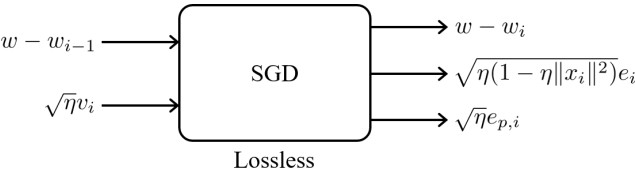

Figure 1: Illustration of Lemma 1. Each step of SGD can be viewed as a transformation of the uncertainties with the right coefficients.

**Lemma 1.** *For any parameter $w$ and noise values $\{v_i\}$ that satisfy $y_i = x_i^T w + v_i$ for $i = 1, \ldots, n$, and for any step size $\eta > 0$, the following relation holds for the SGD iterates $\{w_i\}$ given in Eq. (3)*

$$\|w - w_{i-1}\|^2 + \eta v_i^2 = \|w - w_i\|^2 + \eta \left(1 - \eta \|x_i\|^2\right) e_i^2 + \eta e_{p,i}^2, \quad \forall i \geq 1. \tag{5}$$

As illustrated in Figure 1, this means that each step of SGD can be thought of as a lossless transformation of the input uncertainties to the output uncertainties, with the specified coefficients.

Once one knows this result, proving it is straightforward. To see that, note that we can write $v_i = y_i - x_i^T w$ as $v_i = (y_i - x_i^T w_{i-1}) - (x_i^T w - x_i^T w_{i-1})$. Multiplying both sides by $\sqrt{\eta}$, we have

$$\sqrt{\eta} v_i = \sqrt{\eta}(y_i - x_i^T w_{i-1}) - \sqrt{\eta}(x_i^T w - x_i^T w_{i-1}). \tag{6}$$

On the other hand, subtracting both sides of the update rule (3) from $w$ yields

$$w - w_i = (w - w_{i-1}) - \eta \left(y_i - x_i^T w_{i-1}\right) x_i. \tag{7}$$

Squaring both sides of (6) and (7), and subtracting the results leads to Equation (5).

A nice property of Equation (5) is that, if we sum over all $i = 1, \ldots, T$, the terms $\|w - w_i\|^2$ and $\|w - w_{i-1}\|^2$ on different sides cancel out telescopically, leading to the following important lemma.

**Lemma 2.** *For any parameter $w$ and noise values $\{v_i\}$ that satisfy $y_i = x_i^T w + v_i$ for $i = 1, \ldots, n$, any initialization $w_0$, any step size $\eta > 0$, and any number of steps $T \geq 1$, the following relation holds for the SGD iterates $\{w_i\}$ given in Eq. (3)*

$$\|w - w_0\|^2 + \eta \sum_{i=1}^{T} v_i^2 = \|w - w_T\|^2 + \eta \sum_{i=1}^{T} \left(1 - \eta \|x_i\|^2\right) e_i^2 + \eta \sum_{i=1}^{T} e_{p,i}^2. \tag{8}$$

As we will show next, this identity captures most properties of SGD, and implies several important results in a very transparent fashion. For this reason, this relation can be viewed as a "fundamental identity" for SGD.

### 3.2 Minimax Optimality of SGD

For a given horizon $T$, consider the following minimax problem:

$$\min_{\{w_i\}} \max_{w, \{v_i\}} \frac{\|w - w_T\|^2 + \eta \sum_{i=1}^{T} e_{p,i}^2}{\|w - w_0\|^2 + \eta \sum_{i=1}^{T} v_i^2}. \tag{9}$$

This minimax problem is motivated by the theory of $H^\infty$ control and estimation (Francis, 1987; Hassibi et al., 1999; Başar & Bernhard, 2008). The denominator of the cost function can be interpreted as the *energy of the uncertainties* and consists of two terms, $\|w - w_0\|^2$, the energy of our uncertainty of the unknown weight vector at the beginning of learning when we have not yet observed the data, and $\sum_{i=1}^{T} v_i^2$, the energy of the uncertainty in the measurements. The numerator denotes the energy of the estimation errors in an *online setting*. The first term, $\|w - w_T\|^2$, is the energy of our uncertainty of the unknown weight vector after we have observed $T$ data points, and the second term, $\sum_{i=1}^{T} e_{p,i}^2 = \sum_{i=1}^{T} (x_i^T w - x_i^T w_{i-1})^2$, is the energy of the prediction error, i.e., how well we can predict the true uncorrupted output $x_i^T w$ using measurements up to time $i - 1$. The parameter

$\eta$ weighs the two energy terms relative to each other. In this minimax problem, nature has access to the unknown weight vector $w$ and the noise sequence $v_i$ and would like to maximize the energy gain from the uncertainties to prediction errors (so that the estimator behaves poorly), whereas the estimator attempts to minimize the energy gain. Such an estimator is referred to as $H^\infty$-optimal and is robust because it safeguards against the worst-case noise. It is also conservative—for the exact same reason.[3]

**Theorem 3.** *For any initialization $w_0$, any step size $0 < \eta \le \min_i \frac{1}{\|x_i\|^2}$, and any number of steps $T \ge 1$, the stochastic gradient descent iterates $\{w_i\}$ given in Eq. (3) are the optimal solution to the minimax problem (9). Furthermore, the optimal minimax value (achieved by SGD) is 1.*

This theorem explains the observed robustness and conservatism of SGD. Despite the conservativeness of safeguarding against the worst-case disturbance, this choice may actually be the rational thing to do in situations where we do not have much knowledge about the disturbances, which is the case in many machine learning tasks.

Theorem 3 holds for any horizon $T \ge 1$. A variation of this result, i.e., when $T \to \infty$ and without the $\|w - w_T\|^2$ term in the numerator, was first shown in (Hassibi et al., 1994; 1996). In that case, the ratio $\frac{\eta \sum_{i=1}^\infty e_{p,i}^2}{\|w - w_0\|^2 + \eta \sum_{i=1}^\infty v_i^2}$ in the minimax problem is in fact the $H^\infty$ *norm* of the transfer operator that maps the unknown disturbances $(w - w_0, \{\sqrt{\eta} v_i\})$ to the prediction errors $\{\sqrt{\eta} e_{p,i}\}$.

We end this section with a stochastic interpretation of SGD (Hassibi et al., 1996). Assume that the true weight vector has a normal distribution with mean $w_0$ and covariance matrix $\eta I$, and that the noise $v_i$ are iid standard normal. Then SGD solves

$$\min_{\{w_i\}} \mathbb{E} \exp \left( \frac{1}{2} \cdot \left( \|w - w_T\|^2 + \eta \sum_{i=1}^T (x_i^T w - x_i^T w_{i-1})^2 \right) \right), \qquad (10)$$

and no exponent larger than $\frac{1}{2}$ is possible, in the sense that no estimator can keep the expected cost finite. This means that, in the Gaussian setting, SGD minimizes the expected value of an *exponential* quadratic cost. The algorithm is thus very adverse to large estimation errors, as they are penalized exponentially larger than moderate ones.

### 3.3 CONVERGENCE AND IMPLICIT REGULARIZATION

The over-parameterized (interpolating) linear regression regime is a simple but instructive setting, recently considered in some papers (Gunasekar et al., 2018a; Zhang et al., 2016). In this setting, we can show that, for sufficiently small step, i.e. $0 < \eta \le \min_i \frac{1}{\|x_i\|^2}$, SGD always converges to a special solution among all the solutions $\mathcal{W}$, in particular to the one with the smallest $l_2$ distance from $w_0$. In other words, if, for example, initialized at zero, SGD implicitly regularizes the solution according to an $l_2$ norm. This result follows directly from Lemma 2.

To see that, note that in the interpolating case the $v_i$ are zero, and we have $e_i = y_i - x_i^T w_{i-1} = x_i^T w - x_i^T w_{i-1} = e_{p,i}$. Hence, identity (8) reduces to

$$\|w - w_0\|^2 = \|w - w_T\|^2 + \eta \sum_{i=1}^T \left( 2 - \eta \|x_i\|^2 \right) e_i^2, \qquad (11)$$

for all $w \in \mathcal{W}$. By dropping the $\|w - w_T\|^2$ term and taking $T \to \infty$, we have $\eta \sum_{i=1}^\infty \left( 2 - \eta \|x_i\|^2 \right) e_i^2 \le \|w - w_0\|^2$, which implies that, for $0 < \eta < \min_i \frac{2}{\|x_i\|^2}$, we must have $e_i \to 0$ as $i \to \infty$. When $e_i = y_i - x_i^T w_{i-1}$ goes to zero, the updates in (3) vanish and we get convergence, i.e., $w \to w_\infty$. Further, again because $e_i \to 0$, all the data points are being fit, which means $w_\infty \in \mathcal{W}$. Moreover, it is again very straightforward to see from (11) that the solution converged to is the one with minimum Euclidean norm from the initial point. To see that, notice that

---

[3]The setting described is somewhat similar to the setting of online learning, where one considers the relative performance of an online learner who needs to predict, compared to a clairvoyant one who has access to the entire data set (Shalev-Shwartz, 2012; Hazan, 2016). In online learning, the relative performance is described as a difference, rather than as a ratio in $H^\infty$ theory, and is referred to as *regret*.

the summation term in Eq. (11) is *independent of* $w$ (it depends only on $x_i, y_i$ and $w_0$). Therefore, by taking $T \to \infty$ and minimizing both sides with respect to $w \in \mathcal{W}$, we get

$$w_\infty = \arg\min_{w \in \mathcal{W}} \|w - w_0\|. \tag{12}$$

Once again, this also implies that if SGD is initialized at the origin, i.e., $w_0 = 0$, then it converges to the minimum-$l_2$-norm solution, among all the solutions.

# 4 MAIN RESULT: GENERAL CHARACTERIZATION OF STOCHASTIC MIRROR DESCENT

Stochastic Mirror Descent (SMD) (Nemirovskii et al., 1983; Beck & Teboulle, 2003; Cesa-Bianchi et al., 2012; Zhou et al., 2017) is one of the most widely used families of algorithms for stochastic optimization, which includes SGD as a special case. In this section, we provide a characterization of the behavior of general SMD, on *general* loss functions and *general* nonlinear models, in terms of a fundamental identity and minimax optimality.

For any strictly convex and differentiable potential $\psi(\cdot)$, the corresponding SMD updates are defined as

$$w_i = \arg\min_w \ \eta w^T \nabla L_i(w_{i-1}) + D_\psi(w, w_{i-1}), \tag{13}$$

where

$$D_\psi(w, w_{i-1}) = \psi(w) - \psi(w_{i-1}) - \nabla\psi(w_{i-1})^T(w - w_{i-1}) \tag{14}$$

is the Bregman divergence with respect to the potential function $\psi(\cdot)$. Note that $D_\psi(\cdot, \cdot)$ is non-negative, convex in its first argument, and that, due to strict convexity, $D_\psi(w, w') = 0$ iff $w = w'$. Moreover, the updates can be equivalently written as

$$\nabla\psi(w_i) = \nabla\psi(w_{i-1}) - \eta \nabla L_i(w_{i-1}), \tag{15}$$

which are uniquely defined because of the invertibility of $\nabla\psi$ (again, implied by the strict convexity of $\psi(\cdot)$). In other words, stochastic mirror descent can be thought of as transforming the variable $w$, with a *mirror map* $\nabla\psi(\cdot)$, and performing the SGD update on the new variable. For this reason, $\nabla\psi(w)$ is often referred to as the *dual* variable, while $w$ is the *primal* variable.

Different choices of the potential function $\psi(\cdot)$ yield different optimization algorithms, which, as we will see, result in different implicit regularizations. To name a few examples: For the potential function $\psi(w) = \frac{1}{2}\|w\|^2$, the Bregman divergence is $D_\psi(w, w') = \frac{1}{2}\|w - w'\|^2$, and the update rule reduces to that of SGD. For $\psi(w) = \sum_j w_j \log w_j$, the Bregman divergence becomes the unnormalized relative entropy (Kullback-Leibler divergence) $D_\psi(w, w') = \sum_j w_j \log \frac{w_j}{w_j'} - \sum_j w_j + \sum_j w_j'$, which corresponds to the exponentiated gradient descent (aka the exponential weights) algorithm. Other examples include $\psi(w) = \frac{1}{2}\|w\|_Q^2 = \frac{1}{2}w^T Q w$ for a positive definite matrix $Q$, which yields $D_\psi(w, w') = \frac{1}{2}(w - w')^T Q(w - w')$, and the $q$-norm squared $\psi(w) = \frac{1}{2}\|w\|_q^2$, which with $\frac{1}{p} + \frac{1}{q} = 1$ yields the $p$-norm algorithms (Grove et al., 2001; Gentile, 2003).

In order to derive an equivalent "conservation law" for SMD, similar to the identity (5), we first need to define a new measure for the difference between the parameter vectors $w$ and $w'$ according to the loss function $L_i(\cdot)$. To that end, let us define

$$D_{L_i}(w, w') := L_i(w) - L_i(w') - \nabla L_i(w')^T(w - w'), \tag{16}$$

which is defined in a similar way to a Bregman divergence for the loss function.[4] The difference though is that, unlike the potential function of the Bregman divergence, the loss function $L_i(\cdot) = \ell(y_i - f(x_i, \cdot))$ need not be convex, even when $\ell(\cdot)$ is, due to the nonlinearity of $f(\cdot, \cdot)$. As a result, $D_{L_i}(w, w')$ is not necessarily non-negative. The following result, which is the general counterpart of Lemma 1, states the identity that characterizes SMD updates in the general setting.

**Lemma 4.** *For any (nonlinear) model $f(\cdot, \cdot)$, any differentiable loss $l(\cdot)$, any parameter $w$ and noise values $\{v_i\}$ that satisfy $y_i = f(x_i, w) + v_i$ for $i = 1, \ldots, n$, and any step size $\eta > 0$, the following relation holds for the SMD iterates $\{w_i\}$ given in Eq. (15)*

$$D_\psi(w, w_{i-1}) + \eta l(v_i) = D_\psi(w, w_i) + E_i(w_i, w_{i-1}) + \eta D_{L_i}(w, w_{i-1}), \tag{17}$$

---

[4]It is easy to verify that for linear models and quadratic loss we obtain $D_{L_i}(w, w') = (x_i^T w - x_i^T w')^2$.

*for all $i \geq 1$, where*

$$E_i(w_i, w_{i-1}) := D_\psi(w_i, w_{i-1}) - \eta D_{L_i}(w_i, w_{i-1}) + \eta L_i(w_i). \tag{18}$$

The proof is provided in Appendix A. Note that $E_i(w_i, w_{i-1})$ is not a function of $w$. Furthermore, even though it does not have to be nonnegative in general, for $\eta$ sufficiently small, it becomes nonnegative, because the Bregman divergence $D_\psi(., .)$ is nonnegative.

Summing Equation (17) over all $i = 1, \ldots, T$ leads to the following identity, which is the general counterpart of Lemma 2.

**Lemma 5.** *For any (nonlinear) model $f(\cdot, \cdot)$, any differentiable loss $l(\cdot)$, any parameter $w$ and noise values $\{v_i\}$ that satisfy $y_i = f(x_i, w) + v_i$ for $i = 1, \ldots, n$, any initialization $w_0$, any step size $\eta > 0$, and any number of steps $T \geq 1$, the following relation holds for the SMD iterates $\{w_i\}$ given in Eq. (15)*

$$D_\psi(w, w_0) + \eta \sum_{i=1}^{T} l(v_i) = D_\psi(w, w_T) + \sum_{i=1}^{T} \left( E_i(w_i, w_{i-1}) + \eta D_{L_i}(w, w_{i-1}) \right). \tag{19}$$

We should reiterate that Lemma 5 is a fundamental property of SMD, which allows one to prove many important results, in a direct way.

In particular, in this setting, we can show that SMD is minimax optimal in a manner that generalizes Theorem 3 of Section 3, in the following 3 ways: 1) General potential $\psi(\cdot)$, 2) General model $f(\cdot, \cdot)$, and 3) General loss function $l(\cdot)$. The result is as follows.

**Theorem 6.** *Consider any (nonlinear) model $f(\cdot, \cdot)$, any non-negative differentiable loss $l(\cdot)$ with the property $l(0) = l'(0) = 0$, and any initialization $w_0$. For sufficiently small step size, i.e., for any $\eta > 0$ for which $\psi(w) - \eta L_i(w)$ is convex for all $i$, and for any number of steps $T \geq 1$, the SMD iterates $\{w_i\}$ given by Eq. (15), w.r.t. any strictly convex potential $\psi(\cdot)$, is the optimal solution to the following minimization problem*

$$\min_{\{w_i\}} \max_{w, \{v_i\}} \frac{D_\psi(w, w_T) + \eta \sum_{i=1}^{T} D_{L_i}(w, w_{i-1})}{D_\psi(w, w_0) + \eta \sum_{i=1}^{T} l(v_i)}. \tag{20}$$

*Furthermore, the optimal value (achieved by SMD) is $1$.*

The proof is provided in Appendix B. For the case of square loss and a linear model, the result reduces to the following form.

**Corollary 7.** *For $L_i(w) = \frac{1}{2}(y_i - x_i^T w)^2$, for any initialization $w_0$, any sufficiently small step size, i.e., $0 < \eta \leq \frac{\alpha}{\|x_i\|^2}$, and any number of steps $T \geq 1$, the SMD iterates $\{w_i\}$ given by Eq. (15), w.r.t. any $\alpha$-strongly convex potential $\psi(\cdot)$, is the optimal solution to*

$$\min_{\{w_i\}} \max_{w, \{v_i\}} \frac{D_\psi(w, w_T) + \frac{\eta}{2} \sum_{i=1}^{T} e_{p,i}^2}{D_\psi(w, w_0) + \frac{\eta}{2} \sum_{i=1}^{T} v_i^2}. \tag{21}$$

*The optimal value (achieved by SMD) is $1$.*

We should remark that Theorem 6 and Corollary 7 generalize several known results in the literature. In particular, as mentioned in Section 3, the result of (Hassibi et al., 1994) is a special case of Corollary 7 for $\psi(w) = \frac{1}{2}\|w\|^2$. Furthermore, our result generalizes the result of (Kivinen et al., 2006), which is the special case for the $p$-norm algorithms, again, with square loss and a linear model. Another interesting connection to the literature is that it was shown in (Hassibi & Kailath, 1995) that SGD is *locally* minimax optimal, with respect to the $H^\infty$ norm. Strictly speaking, our result is not a generalization of that result; however, Theorem 6 can be interpreted as SGD/SMD being *globally* minimax optimal, but with respect to different metrics in the numerator and denominator. Namely, the uncertainty about the weight vector $w$ is measured by the Bregman divergence of the potential, the uncertainty about the noise by the loss, and the prediction error by the "Bregman-divergence-like" expression of the loss.

# 5 CONVERGENCE AND IMPLICIT REGULARIZATION IN OVER-PARAMETERIZED MODELS

In this section, we show some of the implications of the theory developed in the previous section. In particular, we show convergence and implicit regularization, in the over-parameterized (so-called interpolating) regime[5], for general SMD algorithms. We first consider the linear interpolating case, which has been studied in the literature, and show that the known results follow naturally from our Lemma 5. Further, we shall obtain some *new* convergence results. Finally, we discuss the implications for nonlinear models, and argue that the same results hold *qualitatively* in highly-overparameterized settings, which is the typical scenario in deep learning.

## 5.1 OVER-PARAMETERIZED LINEAR MODELS

In this setting, the $v_i$ are zero, $\mathcal{W} = \left\{w \mid y_i = x_i^T w, \ i = 1, \ldots, n\right\}$, and $L_i(w) = l(y_i - x_i^T w)$, with any differentiable loss $l(\cdot)$. Therefore, Eq. (19) reduces to

$$D_\psi(w, w_0) = D_\psi(w, w_T) + \sum_{i=1}^T \left(E_i(w_i, w_{i-1}) + \eta D_{L_i}(w, w_{i-1})\right), \qquad (22)$$

for all $w \in \mathcal{W}$, where

$$D_{L_i}(w, w_{i-1}) = L_i(w) - L_i(w_{i-1}) - \nabla L_i(w_{i-1})^T (w - w_{i-1}) \qquad (23)$$

$$= 0 - l(y_i - x_i^T w_{i-1}) + l'(y_i - x_i^T w_{i-1}) x_i^T (w - w_{i-1}) \qquad (24)$$

$$= -l(y_i - x_i^T w_{i-1}) + l'(y_i - x_i^T w_{i-1})(y_i - x_i^T w_{i-1}) \qquad (25)$$

which is notably *independent of $w$*. As a result, we can easily minimize both sides of Eq. (22) with respect to $w \in \mathcal{W}$, which for $T \to \infty$ leads to the following result.

**Proposition 8.** *For any differentiable loss $l(\cdot)$, any initialization $w_0$, and any step size $\eta$, consider the SMD iterates given in Eq. (15) with respect to any strictly convex potential $\psi(\cdot)$. If the iterates converge to a solution $w_\infty \in \mathcal{W}$, then*

$$w_\infty = \arg\min_{w \in \mathcal{W}} D_\psi(w, w_0). \qquad (26)$$

**Remark.** In particular, for the initialization $w_0 = \arg\min_{w \in \mathbb{R}^m} \psi(w)$, if the iterates converge to a solution $w_\infty \in \mathcal{W}$, then

$$w_\infty = \arg\min_{w \in \mathcal{W}} \psi(w). \qquad (27)$$

An equivalent form of Proposition 8 has been shown recently in, e.g., (Gunasekar et al., 2018a).[6] Other implicit regularization results have been shown in (Gunasekar et al., 2018b; Soudry et al., 2017) for classification problems, which are not discussed here. Note that the result of (Gunasekar et al., 2018a) does not say anything about *whether the algorithm converges or not*. However, our fundamental identity of SMD (Lemma 5) allows us to also establish convergence to the regularized point, for some common cases, which will be shown next.

What Proposition 8 says is that depending on the choice of the potential function $\psi(\cdot)$, the optimization algorithm can perform an implicit regularization without any explicit regularization term. In other words, for any desired regularizer, if one chooses a potential function that approximates the regularizer, we can run the optimization without explicit regularization, and if it converges to a solution, the solution must be the one with the minimum potential.

In principle, one can choose the potential function in SMD for *any* desired convex regularization. For example, we can find the maximum entropy solution by taking the potential to be the negative entropy. Another illustrative example follows.

---

[5]In the classical under-parameterized (online streaming) case with white noise, the same theory can be used to establish convergence to the true parameter under the so-called Robbins–Monro conditions ($\sum_{i=1}^\infty \eta_i = \infty, \sum_{i=1}^\infty \eta_i^2 < \infty$) in a very direct and simple way (see Azizan & Hassibi (2019)).

[6]To be precise, the authors in (Gunasekar et al., 2018a) assume convergence to a global minimizer of the loss function $L(w) = \sum_{i=1}^n l(y_i - x_i^T w)$, which with their assumption of the loss function $l(\cdot)$ having a unique finite root is equivalent to assuming convergence to a point $w_\infty \in \mathcal{W}$.

**Example [Compressed Sensing]:** In compressed sensing, one seeks the sparsest solution to an under-determined (over-parameterized) system of linear equations. The surrogate convex problem one solves is:

$$\begin{align} \min \quad & \|w\|_1 \\ \text{subject to} \quad & y_i = x_i^T w, \quad i = 1, \ldots n \end{align} \tag{28}$$

One cannot choose $\psi(w) = \|w\|_1$, since it is neither differentiable nor strictly convex. However, $\psi(w) = \|w\|_{1+\epsilon}$, for any $\epsilon > 0$, can be used. Figure 4 shows a compressed sensing example, with $n = 50$, $m = 100$, and sparsity $k = 10$. SMD was used with a step size of $\eta = 0.001$ and the potential function was $\psi(\cdot) = \|\cdot\|_{1.1}$. SMD converged to the true sparse solution after around 10,000 iterations. On this example, it was an order of magnitude faster than standard $l_1$ optimization.

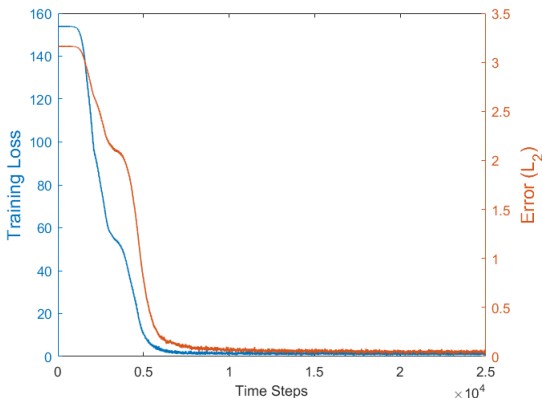

Figure 2: The training loss and actual error of stochastic mirror descent for compressed sensing. SMD recovers the actual sparse signal.

Next we establish *convergence to the regularized point* for the convex case.

**Proposition 9.** *Consider the following two cases.*

(i) *$l(\cdot)$ is differentiable and convex and has a unique root at 0, $\psi(\cdot)$ is strictly convex, and $\eta > 0$ is such that $\psi - \eta L_i$ is convex for all $i$.*

(ii) *$l(\cdot)$ is differentiable and quasi-convex, $l'(\cdot)$ is zero only at zero, $\psi(\cdot)$ is $\alpha$-strongly convex, and $0 < \eta \leq \min_i \frac{\alpha|y_i - x_i^T w_{i-1}|}{\|x_i\|^2 |l'(y_i - x_i^T w_{i-1})|}$.*

*If either (i) or (ii) holds, then for any $w_0$, the SMD iterates given in Eq. (15) converge to*

$$w_\infty = \arg\min_{w \in \mathcal{W}} D_\psi(w, w_0). \tag{29}$$

The proof is provided in Appendix C.

## 5.2 Discussion of Highly Over-Parameterized Nonlinear Models

Let us consider the highly-overparameterized nonlinear model

$$y_i = f(x_i, w), \quad i = 1, \ldots, n, \quad w \in \mathbb{R}^m \tag{30}$$

where by highly-overparameterized we mean $m \gg n$. Since the model is highly over-parameterized, it is assumed that we can perfectly interpolate the data points $(x_i, y_i)$ so that the noise $v_i$ is zero. In this case, the set of parameter vectors that interpolate the data is given by $\mathcal{W} = \{w \in \mathbb{R}^m \mid y_i = f(x_i, w), \; i = 1, \ldots, n\}$, and Eq. (19), again, reduces to

$$D_\psi(w, w_0) = D_\psi(w, w_T) + \sum_{i=1}^{T} \left( E_i(w_i, w_{i-1}) + \eta D_{L_i}(w, w_{i-1}) \right), \tag{31}$$

for all $w \in \mathcal{W}$. Our proofs of convergence and implicit regularization for SGD and SMD in the linear case relied on two facts: (i) $D_{L_i}(w, w_{i-1})$ was non-negative (this allowed us to show convergence), and (ii) $D_{L_i}(w, w_{i-1})$ was independent of $w$ (this allowed us to show implicit regularization). Unfortunately, neither of these hold in the nonlinear case.

However, they do hold in a *local* sense. In other words, (i) $D_{L_i}(w, w_{i-1}) \geq 0$ for $w_{i-1}$ "close enough" to $w$ (see Figure 3), and (ii) $D_{L_i}(w, w_{i-1})$ is weakly dependent on $w$ for $w_{i-1}$ "close enough." (Both statements can be made precise.)

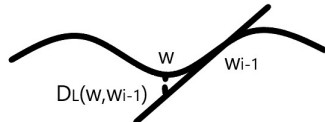

Figure 3: Non-negativity of $D_{L_i}(w, w_{i-1})$ for $w_{i-1}$ "close enough" to $w$.

Now define

$$w_* = \arg \min_{w \in \mathcal{W}} D_\psi(w, w_0). \qquad (32)$$

Then one can show the following result.

**Theorem 10.** *There exists an $\epsilon > 0$, such that if $\|w_* - w_0\| < \epsilon$, then for sufficiently small step size $\eta > 0$:*

1. *SMD iterates converge to a point $w_\infty \in \mathcal{W}$*

2. $\|w_\infty - w_*\| = o(\epsilon)$

This shows that if the initial condition is close enough, then we have convergence to a point $w_\infty$ that interpolates the data, and that $w_\infty$ is an order of magnitude closer to $w_*$ (the implicitly regularized solution) than the initial $w_0$ was. At first glance, this result seems rather dissatisfying. It relies on $w_0$ being close to the manifold $\mathcal{W}$ which appears hard to guarantee. We would now like to argue that in deep learning $w_0$ being close to $\mathcal{W}$ is often the case.

In the highly-overparameterized regime, $m \gg n$, and so the dimension of the manifold $\mathcal{W}$ is $m - n$, which is very large. Now if the $x_i$ are sufficiently random, then the tangent space to $\mathcal{W}$ at $w_*$ will be a randomly oriented affine subspace of dimension $m - n$. This means that any randomly chosen $w_0$ will whp have a very large component when projected onto $\mathcal{W}$. In particular, it can be shown that $\|w_* - w_0\|^2 = O(\frac{n}{m}) \cdot \|y - f(x, w)\|^2$, where $y = \text{vec}(y_i, i = 1, \ldots, n)$ and $f(x, w) = \text{vec}(f(x_i, w), i = 1, \ldots, n)$. Thus, we may expect that, when $m \gg n$, the distance of any randomly chosen $w_0$ to $\mathcal{W}$ will be small and so SMD will converge to a point on $\mathcal{W}$ that approximately performs implicit regularization.

The gist of the argument is that (i) When $m \gg n$, any random initial condition is "close" to the $n - m$ dimensional solution manifold $\mathcal{W}$, (ii) when $w_0$ is "close" to $w_*$, then SMD converges to a point $w_\infty \in \mathcal{W}$, (iii) $w_\infty$ is "an order of magnitude closer" to $w_*$ than $w_0$ was, and (iv) thus, when highly overparamatrized, SMD converges to a point that exhibits implicit regularization.

Of course, this was a very heuristic argument that merits a much more careful analysis. But it is suggestive of the fact that SGD and SMD, when performed on highly-overparameterized nonlinear models, as occurs in deep learning, may exhibit implicit regularization.

## 6 CONCLUDING REMARKS

We should remark that all the results stated throughout the paper extend to the case of time-varying step size $\eta_i$, with minimal modification. In particular, it is easy to show that in this case, the identity (the counterpart of Eq. (19)) becomes

$$D_\psi(w, w_0) + \sum_{i=1}^{T} \eta_i l(v_i) = D_\psi(w, w_T) + \sum_{i=1}^{T} (E_i(w_i, w_{i-1}) + \eta_i D_{L_i}(w, w_{i-1})), \qquad (33)$$

where $E_i(w_i, w_{i-1}) = D_\psi(w_i, w_{i-1}) - \eta_i D_{L_i}(w_i, w_{i-1}) + \eta_i L_i(w_i)$. As a consequence, our main result will be the same as in Theorem 6, with the only difference that the small-step-size condition in this case is the convexity of $\psi(w) - \eta_i L_i(w)$ for all $i$, and the SMD with time-varying step size will be the optimal solution to the following minimax problem

$$\min_{\{w_i\}} \max_{w, \{v_i\}} \frac{D_\psi(w, w_T) + \sum_{i=1}^{T} \eta_i D_{L_i}(w, w_{i-1})}{D_\psi(w, w_0) + \sum_{i=1}^{T} \eta_i l(v_i)}. \tag{34}$$

Similarly, the convergence and implicit regularization results can be proven under the same conditions (See Appendix D for more details on the time-varying case).

This paper opens up a variety of important directions for future work. Most of the analysis developed here is general, in terms of the *model*, the *loss function*, and the *potential function*. Therefore, it would be interesting to study the implications of this theory for specific classes of models (such as different neural networks), specific losses, and specific mirror maps (which induce different regularization biases). Something for future work.

### ACKNOWLEDGMENTS

This work was supported in part by the National Science Foundation under grants CCF-1423663, CCF-1409204 and ECCS-1509977, by a grant from Qualcomm Inc., by NASA's Jet Propulsion Laboratory through the President and Director's Fund, and by Amazon Web Services Inc. and PIMCO LLC through fellowships.

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

# Supplementary Material

## A  PROOF OF LEMMA 4

*Proof.* Let us start by expanding the Bregman divergence $D_\psi(w, w_i)$ based on its definition

$$D_\psi(w, w_i) = \psi(w) - \psi(w_i) - \nabla\psi(w_i)^T(w - w_i).$$

By plugging the SMD update rule $\nabla\psi(w_i) = \nabla\psi(w_{i-1}) - \eta\nabla L_i(w_{i-1})$ into this, we can write it as

$$D_\psi(w, w_i) = \psi(w) - \psi(w_i) - \nabla\psi(w_{i-1})^T(w - w_i) + \eta\nabla L_i(w_{i-1})^T(w - w_i). \quad (35)$$

Using the definition of Bregman divergence for $(w, w_{i-1})$ and $(w_i, w_{i-1})$, i.e., $D_\psi(w, w_{i-1}) = \psi(w) - \psi(w_{i-1}) - \nabla\psi(w_{i-1})^T(w - w_{i-1})$ and $D_\psi(w_i, w_{i-1}) = \psi(w_i) - \psi(w_{i-1}) - \nabla\psi(w_{i-1})^T(w_i - w_{i-1})$, we can express this as

$$\begin{aligned} D_\psi(w, w_i) = D_\psi(w, w_{i-1}) + \psi(w_{i-1}) + \nabla\psi(w_{i-1})^T(w - w_{i-1}) - \psi(w_i) \\ - \nabla\psi(w_{i-1})^T(w - w_i) + \eta\nabla L_i(w_{i-1})^T(w - w_i) \quad (36) \end{aligned}$$

$$\begin{aligned} = D_\psi(w, w_{i-1}) + \psi(w_{i-1}) - \psi(w_i) + \nabla\psi(w_{i-1})^T(w_i - w_{i-1}) \\ + \eta\nabla L_i(w_{i-1})^T(w - w_i) \quad (37) \end{aligned}$$

$$= D_\psi(w, w_{i-1}) - D_\psi(w_i, w_{i-1}) + \eta\nabla L_i(w_{i-1})^T(w - w_i). \quad (38)$$

Expanding the last term using $w - w_i = (w - w_{i-1}) - (w_i - w_{i-1})$, and following the definition of $D_{L_i}(.,.)$ from (16) for $(w, w_{i-1})$ and $(w_i, w_{i-1})$, we have

$$\begin{aligned} D_\psi(w, w_i) = D_\psi(w, w_{i-1}) - D_\psi(w_i, w_{i-1}) + \eta\nabla L_i(w_{i-1})^T(w - w_{i-1}) \\ - \eta\nabla L_i(w_{i-1})^T(w_i - w_{i-1}) \quad (39) \end{aligned}$$

$$\begin{aligned} = D_\psi(w, w_{i-1}) - D_\psi(w_i, w_{i-1}) + \eta\left(L_i(w) - L_i(w_{i-1}) - D_{L_i}(w, w_{i-1})\right) \\ - \eta\left(L_i(w_i) - L_i(w_{i-1}) - D_{L_i}(w_i, w_{i-1})\right) \quad (40) \end{aligned}$$

$$\begin{aligned} = D_\psi(w, w_{i-1}) - D_\psi(w_i, w_{i-1}) + \eta\left(L_i(w) - D_{L_i}(w, w_{i-1})\right) \\ - \eta\left(L_i(w_i) - D_{L_i}(w_i, w_{i-1})\right) \quad (41) \end{aligned}$$

Defining $E_i(w_i, w_{i-1}) := D_\psi(w_i, w_{i-1}) - \eta D_{L_i}(w_i, w_{i-1}) + \eta L_i(w_i)$, we can write the above equality as

$$D_\psi(w, w_i) = D_\psi(w, w_{i-1}) - E_i(w_i, w_{i-1}) + \eta\left(L_i(w) - D_{L_i}(w, w_{i-1})\right). \quad (42)$$

Notice that for any model class with additive noise, and any loss function $L_i$ that depends only on the residual (i.e. the difference between the prediction and the true label), the term $L_i(w)$ depends only on the noise term, for any "true" parameter $w$. In other words, for all $w$ that satisfy $y_i = f(x_i, w) + v_i$, we have $L_i(w) = l(y_i - f(x_i, w)) = l(y_i - (y_i - v_i)) = l(v_i)$. Finally, reordering the terms leads to

$$D_\psi(w, w_i) + \eta D_{L_i}(w, w_{i-1}) + E_i(w_i, w_{i-1}) = D_\psi(w, w_{i-1}) + \eta l(v_i), \quad (43)$$

which concludes the proof. $\qquad\square$

## B  PROOF OF THEOREM 6

*Proof.* We prove the theorem in two parts. First, we show that the value of the minimax is at least 1. Then we prove that the values is at most 1, and is achieved by stochastic mirror descent for small enough step size.

1. Consider the maximization problem

$$\max_{w, \{v_i\}} \frac{D_\psi(w, w_T) + \eta\sum_{i=1}^{T} D_{L_i}(w, w_{i-1})}{D_\psi(w, w_0) + \eta\sum_{i=1}^{T} l(v_i)}.$$

Clearly, the optimal solution(s) and the optimal value of this problem can, and will, be a function of $\{w_i\}$. Similarly, we can also choose feasible points that depend on $\{w_i\}$. Any choice of a feasible point $(\hat{w}, \{\hat{v}_i\})$ gives a lower bound on the value of the problem. Before choosing a feasible point, let us first expand the $D_{L_i}(w, w_{i-1})$ term in the numerator, according to its definition.

$$D_{L_i}(w, w_{i-1}) = l(v_i) - l(y_i - f_i(w_{i-1})) + l'(y_i - f_i(w_{i-1}))\nabla f(w_{i-1})^T(w - w_{i-1}), \quad (44)$$

where we have used the fact that $l(y_i - f_i(w)) = l(v_i)$ for all consistent $w$, in the first term.

Now, we choose a feasible point as follows

$$\hat{v}_i = f_i(w_{i-1}) - f_i(\hat{w}), \quad (45)$$

where $\hat{w}$ is the choice of $w$, as will be described soon. The reason for choosing this value for the noise is that it "fools" the estimator by making its loss on the corresponding data point zero. In other words, for this choice, we have

$$D_{L_i}(w, w_{i-1}) = l(\hat{v}_i) - l(0) + l'(0)\nabla f(w_{i-1})^T(\hat{w} - w_{i-1})$$
$$= l(\hat{v}_i)$$

because $l(0) = l'(0) = 0$. It should be clear at this point that this choice makes the second terms in the numerator and the denominator equal, independent of the choice of $\hat{w}$. What remains to do, in order to show the 1 lower-bound, is to take care of the other two terms, i.e., $D_\psi(w, w_T)$ and $D_\psi(w, w_0)$. As we would like to make the ratio equal to one, we would like to have $D_\psi(w, w_T) = D_\psi(w, w_0)$, which is equivalent to having

$$\psi(w) - \psi(w_T) - \nabla\psi(w_T)^T(w - w_T) = \psi(w) - \psi(w_0) - \nabla\psi(w_0)^T(w - w_0)$$

which is, in turn, equivalent to

$$(\nabla\psi(w_T) - \nabla\psi(w_0))^T w = -\psi(w_T) + \psi(w_0) + \nabla\psi(w_T)^T w_T - \nabla\psi(w_0)^T w_0. \quad (46)$$

Since $\nabla\psi$ is an invertible function, $\nabla\psi(w_T) - \nabla\psi(w_0) \neq 0$, if $w_T \neq w_0$. Therefore, the above equation has a solution for $w$, if $w_T \neq w_0$. As a result, choosing $\hat{w}$ to be a solution to (46) makes $D_\psi(\hat{w}, w_T) = D_\psi(\hat{w}, w_0)$, if $w_T \neq w_0$. For the case when $w_T = w_0$, it is trivial that $D_\psi(\hat{w}, w_T) = D_\psi(\hat{w}, w_0)$ for any choice of $\hat{w}$. In this case, we only need to choose $\hat{w}$ to be different from $w_0$, to avoid making the ratio $\frac{0}{0}$. Hence, we have the following choice

$$\hat{w} = \begin{cases} \text{a solution of (46)} & \text{for } w_T \neq w_0 \\ w_0 + \delta w \text{ for some } \delta w \neq 0 & \text{for } w_T = w_0 \end{cases} \quad (47)$$

Choosing the feasible point $\hat{w}, \{v_i\}$ according to (47) and (45) leads to

$$\max_{w,\{v_i\}} \frac{D_\psi(w, w_T) + \eta\sum_{i=1}^T D_{L_i}(w, w_{i-1})}{D_\psi(w, w_0) + \eta\sum_{i=1}^T l(v_i)}$$
$$\geq \frac{D_\psi(\hat{w}, w_T) + \eta\sum_{i=1}^T l(f_i(w_{i-1}) - f_i(\hat{w}))}{D_\psi(\hat{w}, w_0) + \eta\sum_{i=1}^T l(f_i(w_{i-1}) - f_i(\hat{w}))}. \quad (48)$$

Taking the minimum of both sides with respect to $\{w_i\}$, we have

$$\min_{\{w_i\}} \max_{w,\{v_i\}} \frac{D_\psi(w, w_T) + \eta\sum_{i=1}^T D_{L_i}(w, w_{i-1})}{D_\psi(w, w_0) + \eta\sum_{i=1}^T l(v_i)}$$
$$\geq \min_{\{w_i\}} \frac{D_\psi(\hat{w}, w_T) + \eta\sum_{i=1}^T l(f_i(w_{i-1}) - f_i(\hat{w}))}{D_\psi(\hat{w}, w_0) + \eta\sum_{i=1}^T l(f_i(w_{i-1}) - f_i(\hat{w}))} = 1. \quad (49)$$

The equality to 1 comes from the fact the that the optimal solution of the minimization either has $w_T^* = w_0$ or $w_T^* \neq w_0$, and in both cases the ratio is equal to 1.

2. Now we prove that, under the small step size condition (convexity of $\psi(w) - \eta L_i(w)$ for all $i$), SMD makes the minimax value at most 1, which means that it is indeed an optimal solution. Recall from Lemma 5 that

$$D_\psi(w, w_0) + \eta \sum_{i=1}^T l(v_i) = D_\psi(w, w_T) + \sum_{i=1}^T E_i(w_i, w_{i-1}) + \eta \sum_{i=1}^T D_{L_i}(w, w_{i-1}),$$

where

$$E_i(w_i, w_{i-1}) = D_\psi(w_i, w_{i-1}) - \eta D_{L_i}(w_i, w_{i-1}) + \eta L_i(w_i).$$

It is easy to check that when $\psi(w) - \eta L_i(w)$ is convex, $D_\psi(w_i, w_{i-1}) - \eta D_{L_i}(w_i, w_{i-1})$ is in fact a Bregman divergence (i.e. the Bregman divergence with respect to the potential $\psi(w) - \eta L_i(w)$), and therefore it is nonnegative for any $w_i$ and $w_{i-1}$. Furthermore, we know that the loss $L_i(w_i)$ is also nonnegative for all $w_i$. It follows that $E_i(w_i, w_{i-1})$ is nonnegative for all values of $w_i, w_{i-1}$ and $i$. As a result, we have the following bound.

$$D_\psi(w, w_0) + \eta \sum_{i=1}^T l(v_i) \geq D_\psi(w, w_T) + \eta \sum_{i=1}^T D_{L_i}(w, w_{i-1}). \tag{50}$$

Since the Bregman divergence $D_\psi(w, w_0)$ and the loss $l(v_i)$ are nonnegative, the left-hand side expression is nonnegative, and it follows that

$$\frac{D_\psi(w, w_T) + \eta \sum_{i=1}^T D_{L_i}(w, w_{i-1})}{D_\psi(w, w_0) + \eta \sum_{i=1}^T l(v_i)} \leq 1. \tag{51}$$

In fact, this means that independent of the choice of the maximizer (i.e. for all $\{v_i\}$ and $w$), as long as the step size condition is met, SMD makes the ratio less than or equal to 1.

Combining the results of 1 and 2 above concludes the proof. $\qquad\square$

### B.1 PROOF OF THEOREM 3

*Proof.* This result is a special case of Theorem 6, which was proven above. In this case, $\psi(w) = \frac{1}{2}\|w\|^2$, $f(x_i, w) = x_i^T w$, and $l(z) = \frac{1}{2}z^2$. Therefore, $D_\psi(w, w_T) = \frac{1}{2}\|w - w_T\|^2$, $D_\psi(w, w_0) = \frac{1}{2}\|w - w_0\|^2$, $D_{L_i}(w, w_{i-1}) = \frac{1}{2}(x_i^T w - x_i^T w_{i-1})^2$, and $l(v_i) = \frac{1}{2}v_i^2$, which leads to the result. $\quad\square$

## C PROOF OF PROPOSITION 9

*Proof.* To prove convergence, we appeal again to Equation (22), i.e.

$$D_\psi(w, w_0) = D_\psi(w, w_T) + \sum_{i=1}^T \left( E_i(w_i, w_{i-1}) + \eta D_{L_i}(w, w_{i-1}) \right), \tag{52}$$

for all $w \in \mathcal{W}$. We prove the two cases separately.

1. The proof of case (i) is straightforward. When $l(\cdot)$ is differentiable and convex, $L_i$ is also convex, and therefore $D_{L_i}(w, w_{i-1})$ is nonnegative. Moreover, when $\psi - \eta L_i$ is convex, $E_i(w_i, w_{i-1})$ is also nonnegative. Therefore, the entire summand in Eq. (52) is nonnegative, and has to go to zero for $i \to \infty$. That is because as $T \to \infty$, the sum should remain bounded, i.e., $\sum_{i=1}^\infty \left( E_i(w_i, w_{i-1}) + \eta D_{L_i}(w, w_{i-1}) \right) \leq D_\psi(w, w_0)$. As a result of the non-negativity of both terms in the sum, we have both $E_i(w_i, w_{i-1}) \to 0$ and $D_{L_i}(w, w_{i-1}) \to 0$ as $i \to \infty$, the latter of which implies $L_i(w_{i-1}) \to 0$. This implies that the updates in (15) vanish and we get convergence, i.e., $w \to w_\infty$. Further, again because $L_i(w_{i-1}) \to 0$, and 0 is the unique root of $l(\cdot)$, all the data point are being fit, which means $w_\infty \in \mathcal{W}$.

2. To prove case (ii), note that we have

$$D_{L_i}(w, w_{i-1}) = L_i(w) - L_i(w_{i-1}) - \nabla L_i(w_{i-1})^T(w - w_{i-1}) \tag{53}$$

$$= 0 - l(y_i - x_i^T w_{i-1}) + l'(y_i - x_i^T w_{i-1})x_i^T(w - w_{i-1}) \tag{54}$$

$$= -l(y_i - x_i^T w_{i-1}) + l'(y_i - x_i^T w_{i-1})(y_i - x_i^T w_{i-1}), \tag{55}$$

and

$$E_i(w_i, w_{i-1}) = D_\psi(w_i, w_{i-1}) - \eta D_{L_i}(w_i, w_{i-1}) + \eta L_i(w_i) \tag{56}$$

$$= D_\psi(w_i, w_{i-1}) + \eta \left( L_i(w_{i-1}) + \nabla L_i(w_{i-1})^T(w_i - w_{i-1}) \right) \tag{57}$$

$$= D_\psi(w_i, w_{i-1}) + \eta \left( l(y_i - x_i^T w_{i-1}) - l'(y_i - x_i^T w_{i-1})x_i^T(w_i - w_{i-1}) \right). \tag{58}$$

It follows from (55) and (58) that the summand in Equation (52) is

$$E_i(w_i, w_{i-1}) + \eta D_{L_i}(w, w_{i-1}) = D_\psi(w_i, w_{i-1}) + \eta l'(y_i - x_i^T w_{i-1})(y_i - x_i^T w_i). \tag{59}$$

The first term is a Bregman divergence, and is therefore nonnegative. In order to establish convergence, one needs to argue that the second term is nonnegative as well, so that the summand goes to zero as $i \to \infty$. Since $l(\cdot)$ is increasing for positive values and decreasing for negative values, it is enough to show that $y_i - x_i^T w_{i-1}$ and $y_i - x_i^T w_i$ have the same sign, in order to establish nonnegativity. It is not hard to see that if the distance between the two points is less than or equal to the distance of $y_i - x_i^T w_i$ from the origin, then the signs are the same. In other words, if $|(y_i - x_i^T w_i) - (y_i - x_i^T w_{i-1})| = |x_i^T(w_i - w_{i-1})| \leq |y_i - x_i^T w_{i-1}|$, then the sign are the same.

Note that by the definition of $\alpha$-strong convexity of $\psi(\cdot)$, we have

$$(\nabla\psi(w_i) - \nabla\psi(w_{i-1}))^T(w_i - w_{i-1}) \geq \alpha \|w_i - w_{i-1}\|^2, \tag{60}$$

which implies

$$-\eta\nabla L_i(w_{i-1})^T(w_i - w_{i-1}) \geq \alpha \|w_i - w_{i-1}\|^2, \tag{61}$$

by substituting from the SMD update rule. Upper-bounding the left-hand side by $\eta\|\nabla L_i(w_{i-1})\|\|(w_i - w_{i-1})\|$ implies

$$\eta\|\nabla L_i(w_{i-1})\| \geq \alpha \|w_i - w_{i-1}\|. \tag{62}$$

This implies that we have the following bound

$$|x_i^T(w_i - w_{i-1})| \leq \|x_i\|\|w_i - w_{i-1}\| \leq \frac{\eta\|x_i\|\|\nabla L_i(w_{i-1})\|}{\alpha}. \tag{63}$$

It follows that if $\eta \leq \frac{\alpha|y_i - x_i^T w_{i-1}|}{\|x_i\|\|\nabla L_i(w_{i-1})\|}$, for all $i$, then the signs are the same, and the summand in Eq.(52) is indeed nonnegative. This condition can be equivalently expressed as $\eta \leq \frac{\alpha|y_i - x_i^T w_{i-1}|}{\|x_i\|^2|l'(y_i - x_i^T w_{i-1})|}$ for all $i$, or $\eta \leq \min_i \frac{\alpha|y_i - x_i^T w_{i-1}|}{\|x_i\|^2|l'(y_i - x_i^T w_{i-1})|}$, which is the condition in the statement of the proposition.

Now that we have argued that the summand is nonnegative, the convergence to $w_\infty \in \mathcal{W}$ is immediate. The reason is that both $D_\psi(w_i, w_{i-1}) \to 0$ and $l'(y_i - x_i^T w_{i-1})(y_i - x_i^T w_i) \to 0$, as $i \to \infty$. The first one implies convergence to a point $w_\infty$. The second one implies that either $y_i - x_i^T w_{i-1} = 0$ or $y_i - x_i^T w_i = 0$, which, in turn, implies $w_\infty \in \mathcal{W}$.

$\square$

## D  TIME-VARYING STEP-SIZE

The update rule for the stochastic mirror descent with time-varying step size is as follows.

$$w_i = \arg\min_w \ \eta_i w^T \nabla L_i(w_{i-1}) + D_\psi(w, w_{i-1}), \tag{64}$$

which can be equivalently expressed as $\nabla\psi(w_i) = \nabla\psi(w_{i-1}) - \eta_i\nabla L_i(w_{i-1})$, for all $i$. The main results in this case are as follows.

**Lemma 11.** *For any (nonlinear) model $f(\cdot, \cdot)$, any differentiable loss $l(\cdot)$, any parameter $w$ and noise values $\{v_i\}$ that satisfy $y_i = f(x_i, w) + v_i$ for $i = 1, \ldots, n$, any initialization $w_0$, any step size sequence $\{\eta_i\}$, and any number of steps $T \geq 1$, the following relation holds for the SMD iterates $\{w_i\}$ given in Eq. (64)*

$$D_\psi(w, w_0) + \sum_{i=1}^{T} \eta_i l(v_i) = D_\psi(w, w_T) + \sum_{i=1}^{T} \left( E_i(w_i, w_{i-1}) + \eta_i D_{L_i}(w, w_{i-1}) \right), \qquad (65)$$

*Proof.* The proof is straightforward by summing the following equation for all $i = 1, \ldots, T$

$$D_\psi(w, w_{i-1}) + \eta_i l(v_i) = D_\psi(w, w_i) + E_i(w_i, w_{i-1}) + \eta_i D_{L_i}(w, w_{i-1}), \qquad (66)$$

which can be easily shown in the same way as in the proof of Lemma 4 in Appendix A. $\qquad \square$

**Theorem 12.** *Consider any general model $f(\cdot, \cdot)$, and any differentiable loss function $l(\cdot)$ with property $l(0) = l'(0) = 0$. For sufficiently small step size, i.e., for any sequence $\{\eta_i\}$ for which $\psi(w) - \eta_i L_i(w)$ is convex for all $i$, the SMD iterates $\{w_i\}$ given by Eq. (64) are the optimal solution to the following minimization problem*

$$\min_{\{w_i\}} \max_{w, \{v_i\}} \frac{D_\psi(w, w_T) + \sum_{i=1}^{T} \eta_i D_{L_i}(w, w_{i-1})}{D_\psi(w, w_0) + \sum_{i=1}^{T} \eta_i l(v_i)}. \qquad (67)$$

*Furthermore, the optimal value (achieved by SMD) is* 1.

*Proof.* The proof is similar to that of Theorem 6, as presented in Appendix B. The argument for the upper-bound of 1 is exactly the same. For the second part of the proof, we use the previous Lemma. It follows from the convexity of $\psi(w) - \eta_i L_i(w)$ that $E_i(w_i, w_{i-1}) \geq 0$, and as a result we have

$$\frac{D_\psi(w, w_T) + \sum_{i=1}^{T} \eta_i D_{L_i}(w, w_{i-1})}{D_\psi(w, w_0) + \sum_{i=1}^{T} \eta_i l(v_i)} \leq 1 \qquad (68)$$

for SMD updates, which concludes the proof. $\qquad \square$

The convergence and implicit regularization results hold similarly, and can be formally stated as follows.

**Proposition 13.** *Consider the following two cases.*

    *(i) $l(\cdot)$ is differentiable and convex and has a unique root at 0, $\psi(\cdot)$ is strictly convex, and the positive sequence $\{\eta_i\}$ is such that $\psi - \eta_i L_i$ is convex for all $i$.*

    *(ii) $l(\cdot)$ is differentiable and quasi-convex and has zero derivative only at 0, $\psi(\cdot)$ is $\alpha$-strongly convex, and $0 < \eta_i \leq \frac{\alpha |y_i - x_i^T w_{i-1}|}{\|x_i\|^2 |l'(y_i - x_i^T w_{i-1})|}$ for all $i$.*

*If either (i) or (ii) holds, then for any initialization $w_0$, the SMD iterates given in Eq. (64) converge to*

$$w_\infty = \arg \min_{w \in \mathcal{W}} D_\psi(w, w_0). \qquad (69)$$

*Proof.* The proof is similar to that of Proposition 9, as provided in Appendix C. $\qquad \square$

