# OpenReview forum: "Stochastic Gradient/Mirror Descent: Minimax Optimality and Implicit Regularization"
_ICLR.cc/2019/Conference_

### Official Review · AnonReviewer3 · 2018-11-01
**a nice attempt to study implicit regularization of SGD but not sure whether the contribution is sufficient**

**Rating:** 5
**Confidence:** 3

**Review:**

Optimization algorithms such as stochastic gradient descent (SGD) and stochastic mirror descent (SMD) have found wide applications in training deep neural networks. In this paper the authors provide some theoretical studies to understand why SGD/SMD can produce a solution with good generalization performance when applied to high-parameterized models. The authors developed a fundamental identity for SGD with least squares loss function, based on which the minimax optimality of SGD is established, meaning that SGD chooses the best estimator that safeguards against the worst-case disturbance. Implicit regularization of SGD is also established in the interpolating case, meaning that SGD iterates converge to the one with minimal distance to the starting point in the set of models with no errors. Results are then extended to SMD with general loss functions.

Comments:

(1) Several results are extended from existing literature. For example, Lemma 1 and Theorem 3 have analogues in (Hassibi et al. 1996). Proposition 8 is recently derived in (Gunasekar et al., 2018). Therefore, it seems that this paper has some incremental nature. I am not sure whether the contribution is sufficient enough.

(2) The authors say that they show the convergence of SMD in Proposition 9, while (Gunasekar et al., 2018) does not. It seems that the convergence may not be surprising since the interpolating case is considered there.

(3) Implicit regularization is only studied in the over-parameterized case. Is it possible to say something in the general setting with noises?

(4) The discussion on the implicit regularization for over-parameterized case is a bit intuitive and based on strong assumptions, e.g., the first iterate is close to the solution set. It would be more interesting to present a more rigorous analysis with relaxed assumptions.

---

> ### Author Response · Authors · 2018-11-27
> **Response to Reviewer3**
>
> We thank the reviewer for their feedback and acknowledging the positive aspects of our work. Our responses to the reviewer’s comments follow.
>
> >> (1) Several results are extended from existing literature. For example, Lemma 1 and Theorem 3 have analogues in (Hassibi et al. 1996). Proposition 8 is recently derived in (Gunasekar et al., 2018). Therefore, it seems that this paper has some incremental nature. I am not sure whether the contribution is sufficient enough.<<
>
> As mentioned in the paper, our results differ from these results in several aspects.
> The results on the fundamental identity and minimax optimality, e.g. (Hassibi et al. 1996; Kivinen at al., 2006), had never been shown in this generality, i.e., for general potential functions, general loss functions, and general models. In fact, it was not clear how to extend the results. The key insight here is that one needs to consider the Bregman divergence of the loss function.
>
> While an equivalent form of Proposition 8 has been shown in (Gunasekar et al., 2018), we would like to point out that (1) this result naturally follows from our fundamental identity, and (2) our approach readily proves (deterministic) convergence, too. (Gunasekar et al, 2018) just focus on the KKT conditions after convergence has happened. Their approach does not allow the study of convergence.
>
> >> (2) The authors say that they show the convergence of SMD in Proposition 9, while (Gunasekar et al., 2018) does not. It seems that the convergence may not be surprising since the interpolating case is considered there.<<
>
> We cannot comment on whether convergence in the interpolating case is “surprising” or not. What we can comment on is that proving the convergence of SGD with fixed step size, even in the interpolating case, is not trivial. In fact, (Gunasekar et al., 2018) considered the linear interpolating case too; but to the best of our knowledge, there is no result about convergence in their paper. We further give conditions on the loss function (such as convexity, and even quasi-convexity) for SMD to converge in the linear interpolating case.
>
> >> (3) Implicit regularization is only studied in the over-parameterized case. Is it possible to say something in the general setting with noises?<<
>
> The setting when we the model is not over-parametrized and there is noise, is not as simple. As we mention in the paper, when the model is not over-parameterized, SGD (or SMD) with fixed step size cannot converge. Therefore one cannot speak of implicit regularization when convergence does not happen.
>
> Of course, one can get convergence if the step size is allowed to vanish to zero. In this case, convergence is not surprising, since with a vanishing step size one essentially stops updating the solution after a while. What is more interesting is what one converges to. In work that has been submitted to another venue, we have used the same fundamental identity to show that for iid noise, SGD and SMD converge to the “true” parameter vector, provided the vanishing step size satisfies the so-called Robbins-Monro conditions. Our proof is very simple and direct and avoids ergodic averaging or appealing to stochastic differential equations, which is how the customary proofs go.
>
> >> (4) The discussion on the implicit regularization for over-parameterized case is a bit intuitive and based on strong assumptions, e.g., the first iterate is close to the solution set. It would be more interesting to present a more rigorous analysis with relaxed assumptions.<<
>
> While it would be nice---per the reviewer’s suggestion---to be able to prove convergence without the strong assumption that w_0 be close to the solution set, this may be a bit too ambitious and we are not sure how it can be done---or whether the statement is even true. We should reiterate our belief that this assumption is perhaps not too unrealistic in the highly over-parametrized case, because when the parameters are initialized at random around zero, w.h.p., the initial point will be close to the solution set (which is a very high-dimensional manifold). We have significantly expanded our discussions of the highly over-parametrized nonlinear case in Sec 5.2., with the hope of making the arguments more clear, all while acknowledging the fact that they are somewhat heuristic in nature.

---

### Official Review · AnonReviewer2 · 2018-11-03
**Minimax optimality results are proven for SGD and SMD. These results demonstrate implicit regularization properties of these algorithms even when the models are trained without explicit regularization**

**Rating:** 5
**Confidence:** 3

**Review:**

The authors look at SGD, and SMD updates applied to various models and loss functions. They derive a fundamental identity lemma 2 for the case of linear model and squared loss + SGD and in general for non-linear models+ SMD + non squared loss functions. The main results shown are
1. SGD is optimal in a certain sense for squared loss and linear model.
2. SGD always converges to a solution closest to the starting point.
3. SMD when it converges, converges to a point closest to the starting point in the bregman divergence. The convergence of SMD iterates is shown for certain learning scenarios.

Pros: Shows implicit regularization properties for models beyond linear case.
Cons: 1. The notion of optimality is w.r.t. a metric that is pretty non-standard and it was not clear to me as to why the metric is important to study (the ratio metric in eq 9).
2. The result is not very surprising since SMD is pretty much a gradient descent w.r.t a different distance metric.

---

> ### Author Response · Authors · 2018-11-27
> **Response to Reviewer2**
>
> We thank the reviewer for their constructive feedback and for acknowledging the pros of the work. With respect to the two cons mentioned by the reviewer, we would like to make the following points.
>
> >> 1. The notion of optimality is w.r.t. a metric that is pretty non-standard and it was not clear to me as to why the metric is important to study (the ratio metric in eq 9).<<
>
> While the metric in (9) may be unfamiliar to the learning community, it is known in the estimation theory and control literature, and is in fact the H^{\infty} norm (maximum energy gain) of the transfer operator that maps the unknown disturbances to the prediction errors. H^{\infty} theory was developed to allow the design of estimators and controllers that were robust to model and disturbance uncertainty. There are connections to online learning (that have not yet been fully explored) and we remark on this in the footnote of Section 3.2. Furthermore, extending the minimax optimality results of (Hassibi et al 1996) and (Kivinen et al 2006) to general loss functions and nonlinear models had remained open and our paper shows that the correct way to formulate the minimax problem is through the Bregman divergence of the loss. Finally, the minimax optimality results of SGD and SMD can be regarded as the global defining properties of these algorithms. They are usually defined through some local optimization and/or update and it is not clear what they are doing globally---whether they are optimizing anything globally. Our results show what it is that they globally optimize.
>
> >> 2. The result is not very surprising since SMD is pretty much a gradient descent w.r.t a different distance metric.<<
>
> Stochastic mirror descent (SMD) is a popular family of algorithms, which includes stochastic gradient descent (SGD) as a special case (when the potential function is the squared l2 norm), and has been studied in many papers, e.g. (Nemirovskii et al., 1983; Beck & Teboulle, 2003; Cesa-Bianchi et al., 2012; Zhou et al., 2017; Zhang and He, 2018; etc.). While each step of SMD can be viewed as transforming the variable $w$ with a mirror map, to $\nabla\psi(w)$, and adding the instantaneous gradient update to that variable, the updates are NOT the gradient with respect to that new variable, and therefore, it is not “gradient descent w.r.t. a different metric.” In fact, when the step size is very small, one can show that SMD updates the $w$ vector, not by the instantaneous gradient, but rather by the product of the inverse Hessian of the potential and the instantaneous gradient.
>
> Finally, for clarity, we would like to summarize the contributions of this work:
>
> 1. We show that there exists a “fundamental identity” (i.e., a conservation law) which holds for SMD (and SGD) under very general conditions.
>
> 2. Using this identity, we show that, for general nonlinear models and general loss functions, when the step size is sufficiently small, SMD (and SGD) are the optimal solution of a certain minimax filtering problem. This generalizes several results from the robust control theory literature, e.g., (Hassibi et al., 1994; Kivinen at al., 2006.)
>
> 3. We show that many properties recently proven in the literature, such as the “implicit regularization” of SMD (and SGD) in the over-parameterized linear case---when convergence happens---(Gunasekar et al., 2018), naturally follow from this theory. The theory also allows us to establish new results, such as the convergence (in a deterministic sense) of SMD (and SGD) in the over-parameterized linear case.
>
> 4. We finally also use the theory developed in this paper to provide some speculative arguments into why SMD (and SGD) may have similar convergence and implicit regularization properties in the so-called ``highly over-parameterized'' nonlinear setting common to deep learning.

---

> > ### Comment · AnonReviewer2 · 2018-11-28
> > **Response after authors rebuttals.**
> >
> > I thank the authors for having responded to my questions.
> > The description of mirror descent provided by the authors is correct. An alternative description of mirror descent is that it is similar to gradient descent with Bregman divergence induced by the strongly convex potential function. Look at Proposition 3.2 in the following paper which establishes this equivalence
> > https://web.iem.technion.ac.il/images/user-files/becka/papers/3.pdf
> > With this viewpoint, I remarked (in my official review) that implicit regularization property of SMD algorithm is not surprising.
> >
> > While the fundamental identity the authors prove is interesting for robust control community, I as a machine learning researcher, find it hard to appreciate thi result. Modulo these, the contributions are very incremental. For this reason, I cannot recommend a strong acceptance.

---

> > > ### Author Response · Authors · 2018-11-29
> > > **Response**
> > >
> > > We thank the reviewer for their additional comments and have noted that they increased their score. We are not in disagreement with regards to SMD, or the reviewer's clarifying remarks about it. Furthermore, as also mentioned to Reviewer 3, we cannot comment on whether the implicit regularization of SMD is "surprising" or not.
> > >
> > > However, we do regretfully disagree with the reviewer that the paper's contributions are incremental.The reviewer bases this contention on their assertions that the implicit regularization of SMD is not surprising and that, as a machine learning researcher, they cannot appreciate the fundamental identity we show for SMD. We are not sure what to make of this last statement. The fundamental identity we show for SMD---both the local version in Lemma 4 and the global version in Lemma 5---can be regarded as a "defining property" of SMD, in the same way that our Eq (13), or Eqs (3.9) and (3.11) in the reference the reviewer cites, are defining properties of SMD. In other words, the SMD updates can be obtained from the identities in Lemmas 4 and 5, and therefore "define" the SMD updates. The advantage of these lemmas, especially Lemma 5, is that it gives a "global" interpretation of what SMD does, something that is not apparent---at all---from the defining local optimization of Eq (13) or the explicit update (15). It says something about what SMD is doing, and what quantities it is preserving, something which is not directly apparent from (13) or (15). It shows that the sum of D_{Li}(w,w_{i-1}), a certain measure of how well we are predicting the true parameter vector w, is bounded above by the sum of l(v_i), the loss of the noise. For quadratic loss, it upper bounds the energy of the prediction error by the energy of the noise.
> > >
> > > In addition to yielding a novel interpretation for SMD, we show the utility of this fundamental identity, both to derive novel results, as well as to obtain more direct proofs of existing ones. We establish the minimax optimality of SMD, which generalizes the H-infinity optimality of SGD for linear models and quadratic loss. (Perhaps this is what the reviewer contends only the robust control community would appreciate. But, even if that were so---like it or not---this is a property of SMD that no other algorithms possess. It also can be interpreted in terms of the robustness of the algorithm in a manner we describe in the paper.) We further use the fundamental identity to give a deterministic proof of convergence for fixed step-size SMD in the over-parametrized case---something that had not been done before---and re-obtain implicit regularization in a very transparent way. The identity also allows us to say quite a bit in the over-parametrized nonlinear case (as happens in deep learning), and we outline this in Section 5.2. (The nonlinear case is currently under further investigation.) We have also used the fundamental identity to give a very direct proof of the stochastic convergence of SMD when the step size is vanishing and satisfies the Robbins-Monro conditions (this has been submitted to another venue).
> > >
> > > Further, as mentioned by Reviewer 1, our fundamental identity raises the question of whether such identities can be found for other, perhaps more complicated, algorithms.
> > >
> > > All this appears novel to us and we do not know what any of it has to do with being a machine learning researcher. SMD is used in machine learning and, in our view, new facts about it are expected to be of interest to machine learning researchers and practitioners.
> > >
> > > We sincerely appreciate the reviewer's time and efforts in reading and evaluating our paper and value their comments. However, we had hoped the reviewer's recommendation would be based on objective facts, rather than subjective "surprise" and "appreciation".

---

### Official Review · AnonReviewer1 · 2018-11-05
**Very insightful paper but some essential details are missing.**

**Rating:** 7
**Confidence:** 4

**Review:**

This is a very interesting paper and it suggests a novel way to think of "implicit regularization". The power of this paper lies in its simplicity and its inspiring that such almost-easy arguments could be made to get so much insight. It suggests that minimizers of the Bregrman divergence are an alternative characterization of the asymptotic end-points of  "Stochastic Mirror Descent" (SMD) when it converges. So choice of the strongly convex potential function in SMD is itself a regularizer!

Its a very timely paper given the increasing consensus that "implicit regularization" is what drives a lot of deep-learning heuristics. This paper at its technical core suggests a modified notion of Bregman-like divergence (equation 15) which on its own does not need a strongly convex potential. Then the paper goes on to show that there is an invariant of the iterations of SMD along its iterations which involves a certain relationship (equation 18) between the usual Bregman divergence and their modified divergence. I am eager to see if such relationships can be shown to hold for more complicated iterative algorithms!

But there are a few points in the paper which are not clear and probably need more explanation and let me list them here. ( and these are the issues that prevent me from giving this paper a very high rating despite my initial enthusiasm )

1.
Can the authors explain how is the minimax optimality result of Theorem 6 (and Corollary 7) related to the main result of the paper which is probably Proposition 8 and and 9? Is that minimax optimiality a different insight separate from the main line of the arguments (which I believe is Proposition 8 and 9)?

2.
Is the gain in Proposition 9 over Proposition 8 is all about using loss convexity to ensure that the SMD converges and w_\infty exists?

3.
The paper has highly insufficient comparisons to many recent other papers on the idea of "implicit bias" like, https://arxiv.org/abs/1802.08246, https://arxiv.org/abs/1806.00468 and https://arxiv.org/abs/1710.10345. It seems pretty necessary that there be a section making a detailed comparison with these recent papers on similar themes.

---

> ### Author Response · Authors · 2018-11-27
> **Response to Reviewer1**
>
> We thank the reviewer for their supportive feedback and comments. We agree that it would be nice to see whether invariant relationships of the type we have found for SMD were to hold for more complicated iterative algorithms---we are currently investigating this. To the best of our abilities, we have made every attempt to clarify the paper and add more explanations and detailed discussions (as permitted by the page limitation). We hope this removes the barriers to a higher score. Now to the specific comments:
>
> >> 1. Can the authors explain how is the minimax optimality result of Theorem 6 (and Corollary 7) related to the main result of the paper which is probably Proposition 8 and and 9? Is that minimax optimiality a different insight separate from the main line of the arguments (which I believe is Proposition 8 and 9)?<<
>
> Yes, we consider the minimax optimality (Theorem 6) as a separate insight. It gives a new interpretation to SMD and shows the manner in which it is robust to uncertainty about the true parameter vector and the model of the noise sequence. It derives from the same identity (18), and extends known results in the estimation theory literature  (e.g., Hassibi et al.; 1996, Kivinen at al., 2006) to general SMD algorithms with general potential and general loss.
>
> >> 2. Is the gain in Proposition 9 over Proposition 8 is all about using loss convexity to ensure that the SMD converges and w_\infty exists?<<
>
> Yes, that is correct.
>
> >> 3. The paper has highly insufficient comparisons to many recent other papers on the idea of "implicit bias" like, https://arxiv.org/abs/1802.08246, https://arxiv.org/abs/1806.00468 and https://arxiv.org/abs/1710.10345. It seems pretty necessary that there be a section making a detailed comparison with these recent papers on similar themes.<<
>
> Thank you for pointing out the above references---we have added them all. We also provide a brief comparison to our results (see Sec 1.1 “Our Contributions”, as well as the discussion below Proposition 8). The main difference is that our techniques allow a (deterministic) proof of convergence of SMD for the regression problem, which was not given in prior papers (implicit regularization was shown if convergence happens).

---

### Meta-Review · Area_Chair1 · 2018-12-16
**Interesting generalization of older results in control**

**Confidence:** 3
**Recommendation:** Accept (Poster)

**Metareview:**

The authors give a characterization of stochastic mirror descent (SMD) as a conservation law (17) in terms of the Bregman divergence of the loss. The identity allows the authors to show that SMD converges to the optimal solution of a particular minimax filtering problem. In the special overparametrized linear case, when SMD is simply SGD, the result recovers a recent theorem due to Gunasekar et al. (2018). The consequences for the overparametrized nonlinear case are more speculative.

The main criticisms are around impact, however, I'm inclined to think that any new insight on this problem, especially one that imports results from other areas like control, are useful to incorporate into the literature.

I will comment that the discussion of previous work is wholly inadequate. The authors essentially do not engage with previous work, and mostly make throwaway citations. This is a real pity.  I would be nice to see better scholarship.